# Research on government subsidy strategy of biomass power supply chain considering channel encroachment

Xin Wu[1], Peng Liu[2], Jin Li[1], Jing Gao[3], Guangyin Xu [1,4*], Heng Wang[1], Hang Ke[1]

1 College of Mechanical and Electrical Engineering, Henan Agricultural University, Zhengzhou, Henan Province, China, 2 School of Management Engineering, Henan University of Engineering, Zhengzhou, Henan Province, China, 3 School of Logistics and Electronic Commerce, Henan University of Animal Husbandry and Economy, Zhengzhou, Henan Province, China, 4 Institute of Agricultural Engineering, Huanghe S and T University, Zhengzhou, Henan Province, China

* xgy4175@henau.edu.cn

## Abstract

To enhance the comprehensive utilization of biomass straw, governments may implement incentive policies for members of the biomass supply chain. This study examines the strategic interaction between government subsidy strategies and farmers' channel encroachment strategies within the biomass power supply chain. A game-theoretic model is employed to analyze eight government subsidy scenarios, leading to the following conclusions: In the absence of encroachment, subsidies provided to either middlemen or farmers contribute to increased profits for the respective recipients. Notably, the analysis indicates that under the encroachment scenario, government subsidies directed solely to middlemen may negatively affect the overall social welfare of the biomass power generation supply chain. Furthermore, as channel competition intensifies, the probability and extent of this negative impact on social welfare are likely to increase. Additionally, the equilibrium outcome of the game-theoretic model establishes that farmers will invariably choose encroachment as a means to trigger government subsidies, thereby maximizing their profits. These findings provide essential theoretical insights into farmers' strategic behavior aimed at income enhancement and offer guidance for government subsidy policies to achieve optimal social welfare.

## 1. Introduction

As conventional energy sources continue to be depleted and greenhouse gas emissions escalate, the challenges of global environmental pollution and climate change have become increasingly urgent and significant [1]. In response, governments have implemented a series of policies and regulations aimed at achieving energy conservation and emission reduction. During the 75th session of the United Nations General

**Data availability statement:** All relevant data are within the manuscript.

**Funding:** This research was supported by two projects: "Route Planning of Fresh Agricultural Product Distribution Vehicles under Traffic Control Policies" (Henan Province Science and Technology Research Project, 2024, No. 242102240028) and "Research on Risk Evolution and Control Mechanism of Straw-based Agricultural Waste Supply Chain" (Henan Provincial Natural Science Foundation, 2025, No. 252300420852). Professor Wang Heng served as the principal investigator for both projects and is also a co-author of this paper. In this study, he provided methodological guidance and conducted field investigations.

**Competing interests:** The authors have declared that no competing interests exist.

Assembly, China established a "double carbon target," setting goals to peak carbon dioxide emissions by 2030 and attain carbon neutrality by 2060. Similarly, the EU has committed to reducing greenhouse gas emissions by at least 55% from 1990 levels by 2030, with the ultimate objective of reaching carbon neutrality by 2050 [2]. The transition towards renewable energy has become an inevitable trend. Among various renewable energy sources, biomass energy, recognized as the fourth largest global energy resource, has gained prominence as a crucial approach for governments to achieve sustainable energy development due to its "carbon-neutral" characteristic [3]. Biomass energy utilization primarily includes biomass power generation as a key application [4], with its supply chain comprising three main components: farmers responsible for the production and supply of feedstock, middlemen managing the transportation and storage of biomass feedstock, and biomass power plants that utilize biomass for electricity generation [5]. However, the high cost associated with biomass raw material supply poses a constraint on its energy utilization, as studies indicate that biomass supply costs account for about 55% of total power generation expenses [6]. According to data from the 2024 National Straw Comprehensive Utilization Promotion Conference, more than 10% of straw resources in China remain underutilized, with only 8.3% allocated for energy purposes out of the total available resources. The development of the biomass power generation supply chain continues to require government intervention [7]. Furthermore, information asymmetry within the biomass supply chain has reduced suppliers' motivation [8], resulting in inefficiencies in biomass raw material supply channels [9].

Faced with these challenges, countries typically adopt strategies to guide and regulate the decision-making behaviors of biomass supply chain participants through the formulation of relevant policies, thereby promoting the sustainable development of the biomass power supply chain [10]. Comprehensive crop straw subsidies are primarily designed to support biomass supply chain participants, including core enterprises, professional cooperatives, family farms, large professional households, and individual households [11]. These subsidies directly benefit stakeholders within the biomass supply chain. By providing policy incentives, governments seek to encourage relevant entities to engage in the comprehensive utilization of crop straw, thereby improving its utilization rate [12] and enhancing social welfare [13] within the biomass supply chain. Additionally, this policy helps mitigate the adverse effects of straw incineration, addressing the decline in air quality caused by straw burning. The feed-in tariff subsidy policy for biomass power generation, which aims to secure the revenue of enterprises, represents one of the most widely adopted incentive measures [14]. Denmark employs a direct approach by establishing a fixed feed-in tariff for biomass power generation at 4.1 euro cents per kilowatt-hour with a 10-year guarantee [15]. In addition to tariff subsidies, certain governments provide financial support based on the volume of biomass straw utilized [16,17]. For instance, Zongyang County in Anhui Province, China, offers subsidies of RMB 20 per ton for wheat and rice straw and RMB 30 per ton for rape straw, applicable to industrialized straw utilization projects with an annual volume between 50 and 1,000 tons. For straw exceeding 1,000 tons annually, additional subsidies are allocated at the provincial level [18]. Regarding

transportation, the Danish government provides subsidies for biomass fuel supply chains [19] while also implementing a tax exemption policy for electric vehicles powered by biomass fuels [20]. However, in general, no clear and unified regulation exists concerning the recipients and distribution mechanisms of subsidies within the biomass power generation supply chain.

Conversely, in the biomass raw material supply sector, farmers are increasingly choosing to bypass middlemen [21] and supply raw materials directly to power plants [22]. For example, in Gongzhuling City, Jilin Province, following the autumn harvest, farmers transport packaged straw directly to Guoneng Gongzhuling Biomass Power Generation Co., Ltd. [23]. Similarly, in Tengzhou City, Shandong Province, harvested wheat straw is baled and sold directly to the local biomass power plant by farmers [24]. Existing case studies indicate that the expansion of biomass raw material supply channels by farmers may also be influenced by government subsidies. For instance, the "Notice on the Subsidy Funds for the Comprehensive Utilization of Rice Straw in 2022" issued in Qingpu District, Shanghai, specifies that entities purchasing rice straw and implementing straw utilization away from fields within the district are eligible for subsidies of 300 RMB/ton, with the city contributing 240 RMB/ton and the district providing 60 RMB/ton. Additionally, all business units and enterprises involved in the comprehensive utilization management of straw across the province receive a subsidy of 20 RMB per ton based on the quantity of corn and rice straw utilized [25]. These examples highlight the significant role of government subsidies in facilitating the expansion of farmers' supply channels. However, the interaction between government subsidies and channel expansion remains an area that has received limited scholarly attention.

Based on these considerations, this study examines the government's subsidy strategy for members of the biomass straw power supply chain, addressing the following key questions: (1) How does the government subsidy strategy influence the pricing decisions of biomass supply chain participants? (2) Are farmers inclined to expand biomass supply channels (i.e., engage in encroachment), and if so, how does such channel encroachment impact government subsidies? (3) What constitutes the equilibrium outcome of the overall game, and what factors influence this equilibrium outcome?

This study makes several contributions to existing literature. First, to the best of current knowledge, it is the first to investigate government subsidy strategies in biomass power supply chains consisting of farmers and middlemen under horizontal competition. Previous research has primarily focused on competition among vertically related members. Second, this study derives pricing equilibrium strategies for different stakeholders under various government subsidy schemes, yielding several notable findings. Unlike scenarios without encroachment, where subsidies consistently enhance social welfare, subsidizing middlemen in the presence of encroachment may negatively impact social welfare. Furthermore, the findings suggest that under intensified channel competition, a lower subsidy to middlemen increases the probability of a decline in social welfare.

## 2. Literature review

This study investigates the influence of different government subsidy strategies on the decision-making processes of biomass power generation supply chain members, particularly within the competitive dynamics of biomass raw material supply channels. The research primarily focuses on two critical areas: government policies related to the biomass supply chain and the management of raw material supply channels within this sector.

In the academic field, considerable research has been conducted on biomass supply chain policies and subsidy mechanisms. By developing a co-evolution model of the biomass supply chain, Wang et al. [26] demonstrated that an increase in straw prices positively influences the coordinated development of the supply chain and highlighted that higher government subsidies can effectively promote the expansion of straw power generation systems. Ma [27], through computer simulations of the forest biomass power supply chain system, analyzed the impact of various forestry policies on supply chain profitability while also examining the decision-making behavior of key stakeholders. Ghani et al. [28] proposed a decision support system designed to optimize profits while reducing carbon emissions within the biomass supply chain, enabling an assessment of the effects of incentives, emission pricing, and trading mechanisms. Additionally, several studies have

explored the relationship between policy measures and supply chain risks. Liu et al. [29] conducted an in-depth analysis of the challenges currently faced by the biomass power generation industry, identifying potential barriers to future development and conducting a feasibility assessment. The findings suggest that the government should formulate and implement comprehensive support policies across all stages of the supply chain while enhancing the supervision and regulation of biomass projects to effectively mitigate risks. Examining the correlation between government funding and feed-in tariffs for biomass power plants, Li et al. [30] found that when government financial support falls below a critical threshold, implementing technical subsidies for relevant institutions becomes particularly essential.

In existing literature, numerous scholars have applied game theory to analyze the interaction between government decision-making and benefits within the biomass supply chain. Hamed and Reza [31] examined the relationship between biomass material suppliers and mixed-fuel power plants, suggesting that backup fossil fuels can mitigate uncertainties in biomass supply. Their findings also indicate that appropriate government incentives and penalties encourage power plants to adopt biomass materials. He et al. [32] employed an evolutionary game model to investigate government regulations, including administrative penalties and subsidy policies, in the biomass straw supply chain. Their analysis concluded that a dynamic penalty mechanism provides stronger incentives for power plants to utilize biomass energy. When bioenergy constitutes a significant share of a power plant's or farmer's energy mix, government subsidies should be provided if their value exceeds the expected fines; otherwise, punitive measures are more appropriate. Lin et al. [33] categorized biomass subsidies and developed a game model between the government and biomass refineries, concluding that transportation subsidies are more cost-effective in enhancing biomass utilization, while a combination of operational and product subsidies improves industry profitability. Jiang et al. [8] analyzed the optimal government subsidy strategy for a biomass feedstock supply chain involving power plants, village committees, and farmers, identifying stakeholders' best decision-making strategies. Wang et al. [34] used a Stackelberg game model to examine China's biomass supply chain, designing an incentive mechanism based on stakeholder risk perception. Their study found that these incentives not only enhance straw supply but also increase stakeholder profits, with risk perception significantly affecting overall social welfare. Wu et al. [35] constructed and evaluated three scenarios: a non-cooperative game, a cooperative game between farmers and brokers, and a cooperative game between brokers and biomass power plants, all under government incentives. Their findings highlight the government's crucial role in advancing agricultural biomass power generation, particularly in cooperative arrangements. Wang et al. [36] developed a decentralized decision-making model using Stackelberg game theory and a centralized model considering overall revenue. The introduction of a revenue-sharing contract demonstrated that an optimal combination of straw purchase subsidies and revenue-sharing coefficients could achieve Pareto efficiency. Existing research on bioenergy supply chain incentives primarily focuses on vertical competition among supply chain members, often overlooking the impact of channel encroachment on government subsidy decisions. In practice, the channel structure of the biomass power generation supply chain is more complex [37]. Unlike previous studies, this research integrates competition within supply channels and real-world conditions to examine government incentive policies in the biomass power generation supply chain. The findings illustrate how government subsidy strategies influence the equilibrium price for each party in the biomass feedstock supply chain and provide insights into setting optimal subsidy strategies under varying levels of competition across different channels.

The stability of biomass raw material supply is essential for ensuring the reliability and efficiency of the biomass power generation supply chain. Several studies have examined biomass feedstock supply channels and the effects of channel competition. For instance, Tan [38] employed factor analysis to identify key factors influencing farmers' willingness and behavior in biomass resource harvesting, subsequently proposing innovative approaches to enhance biomass harvesting practices. Luo et al. [39] introduced a formal rural organization to expand supply channels, finding that cooperative incentives provided by such organizations positively influenced farmers' willingness to participate. Additionally, policies that promote benefit-sharing significantly impact the equilibrium ratio of farmers engaged in cooperative efforts. Fan et al. [40] developed a biomass supply chain model incorporating farmers, intermediaries, and manufacturers, where farmers decide whether to

supply raw materials to manufacturers or intermediaries, while intermediaries determine their supply decisions to manufacturers. Their research demonstrated that supply chain coordination could be achieved through "protected price plus subsidy" contracts and "buy-back plus revenue sharing" contracts. Sun et al. [41] examined an optimal strategy for a biomass supply chain with competing buyer channels, constructing a supply chain game model consisting of a supplier and two buyers. Their findings indicate that under conditions of low profit cost, both buyers can share agricultural biomass resources efficiently. While these studies highlight the influence of biomass feedstock supply channels on supply chain stability, they do not account for the government's role as an external coordinating force in the biomass supply chain. In practice, regulatory authorities often implement policies designed to influence the behavior of supply chain members, enhance participation incentives, and optimize social welfare. This study focuses on analyzing the impact of government subsidy strategies on decision-making processes among supply chain participants across different biomass feedstock supply channels and levels of channel competition. More importantly, it provides managerial insights for policymakers aiming to regulate and promote the sustainable development of biomass raw material supply chains. A comparison of this study with existing literature is presented in Table 1.

## 3. Model framework

This study examines a biomass supply chain consisting of power plants ($P$), intermediaries ($M$), and farmers ($F$) within a practical biomass supply chain framework. In this system, farmers supply biomass feedstock to intermediaries, who then collect and transport it to power plants. However, to maximize their own profits, farmers may also choose to bypass intermediaries by collecting and selling biomass stalks directly to power plants. To enhance the industrial utilization of straw, improve the efficiency of biomass collection, storage, transportation, and marketing networks, and promote an industrial utilization model driven by biomass power plants, government authorities hold both the responsibility and the authority to formulate policies that regulate and guide the industrialized utilization of straw, including decisions on subsidy allocation. This setup follows a standard Stackelberg game framework, where one party assumes the role

**Table 1. Comparison with the literature.**

| | Incentive policy | | | Channel | | Policy target | | |
|---|---|---|---|---|---|---|---|---|
| | Taxes | Quota system | Subsidization | Encroachment | Non-encroachment | Farmer | Middleman | Manufacturer |
| [26] | | | ✓ | | ✓ | ✓ | | |
| [27] | | | ✓ | ✓ | | | | ✓ |
| [28] | | ✓ | ✓ | | ✓ | ✓ | | ✓ |
| [29] | ✓ | | | | ✓ | | | ✓ |
| [30] | | | ✓ | | ✓ | | | ✓ |
| [31] | | ✓ | ✓ | ✓ | | | ✓ | ✓ |
| [32] | | ✓ | ✓ | | ✓ | ✓ | | ✓ |
| [33] | | | ✓ | | ✓ | | ✓ | ✓ |
| [8] | | ✓ | ✓ | | ✓ | ✓ | ✓ | ✓ |
| [34] | | | ✓ | | ✓ | ✓ | ✓ | |
| [35] | | | ✓ | | ✓ | ✓ | ✓ | ✓ |
| [36] | | | ✓ | | ✓ | ✓ | ✓ | ✓ |
| [38] | | | | ✓ | | | | |
| [39] | | | ✓ | ✓ | | ✓ | | |
| [40] | | | ✓ | | ✓ | ✓ | ✓ | ✓ |
| [41] | | | | ✓ | | | | |
| **This paper** | | | ✓ | ✓ | ✓ | ✓ | ✓ | ✓ |

of a leader and makes an initial decision, while the follower observes this decision before determining its own response [42]. The Stackelberg game is a sequential decision-making model solved through backward induction, widely applied in hierarchical decision-making scenarios [43]. To enhance clarity, the parameter symbols and their respective descriptions are provided in Table 2.

Specifically, this study constructs and evaluates eight decision-making scenarios based on different government policy interventions and channel configurations. The analysis investigates whether farmers choose to bypass intermediaries and supply biomass directly to power plants $(N/E)$ and examines the corresponding government subsidy strategies $(N/F/M/P)$ [44,45].

(1) "Scenario NN Fig 1a": There is No channel competition, and the relevant sector does Not subsidize the supply chain members.

(2) "Scenario NF Fig 1b": There is No channel competition, and the relevant sector subsidizes the farmers within the supply chain.

(3) "Scenario NM Fig 1c": There is No channel competition, and the relevant sector subsidizes the middlemen in the supply chain.

(4) "Scenario NP Fig 1d": There is No channel competition, and the relevant sector subsidizes the power plants in the supply chain.

(5) "Scenario EN Fig 1e": Farmers opt to establish new supply channels, and the relevant sector does Not subsidize the supply chain members.

(6) "Scenario EF Fig 1f"": Farmers opt to establish new supply channels, and the relevant sector subsidizes the farmers in the supply chain.

(7) "Scenario EM Fig 1g": Farmers opt to establish new supply channels, and the relevant sector subsidizes the middlemen in the supply chain.

(8) "Scenario EP Fig 1h": Farmers opt to establish new supply channels, and the relevant sector subsidizes the power plants in the supply chain.

**Table 2. Symbols.**

| Symbols | Connotation |
|---|---|
| $w_1$ | Wholesale prices of farmers selling to middlemen. |
| $p_1$ | The unit price at which the middleman sells to the power plant. |
| $p_2$ | The unit price at which the farmer sells to the power plant. |
| $p_0$ | Unit tariffs for biomass power plants. |
| $t$ | Conversion factor between straw supply and electricity generation. |
| $d_1$ | Demand for channel 1. |
| $d_2$ | Demand for channel 2. |
| $d_1^N$ | Demand for channel 1 in case of non-encroachment. |
| $d_1^E$ | Demand for channel 1 at the time of the encroachment. |
| $c_0, c_1, c_2, c_3$ | Transportation costs at all stages. $c \ 0$ |
| $b$ | The intensity of competition between channels. $1 \ b \ 0$ |
| $a$ | Demand for biomass feedstock. |
| $CS$ | The consumer surplus function. |
| $SW$ | Social welfare. |
| $m$ | The government subsidized unit *prices* for biomass feedstocks. $m \ 0$ |

In addition, the unit processing cost of biomass in different sales channels is denoted as $c_0/c_1/c_2/c_3$ for farmers, middlemen, and power plants, respectively [46]. The demand function [47]:

$$d_1^N = a - p_1,$$ (1)

$$d_1^E = a - p_1 + bp_2,$$ (2)

$$d_2 = a - p_2 + bp_1.$$ (3)

The profit of members in the biomass power generation supply chain is denoted as $\pi_i(a = F; M; P)$, where $F$ represents farmers, $M$ represents middlemen, and $P$ represents power plants. In existing literature, consumer surplus is commonly defined as "the additional benefit or surplus that consumers derive from market transactions above the price paid [48,49]." The consumer surplus function [50]:

$$CS = \frac{(d_1 + d_2)^2}{2}$$ (4)

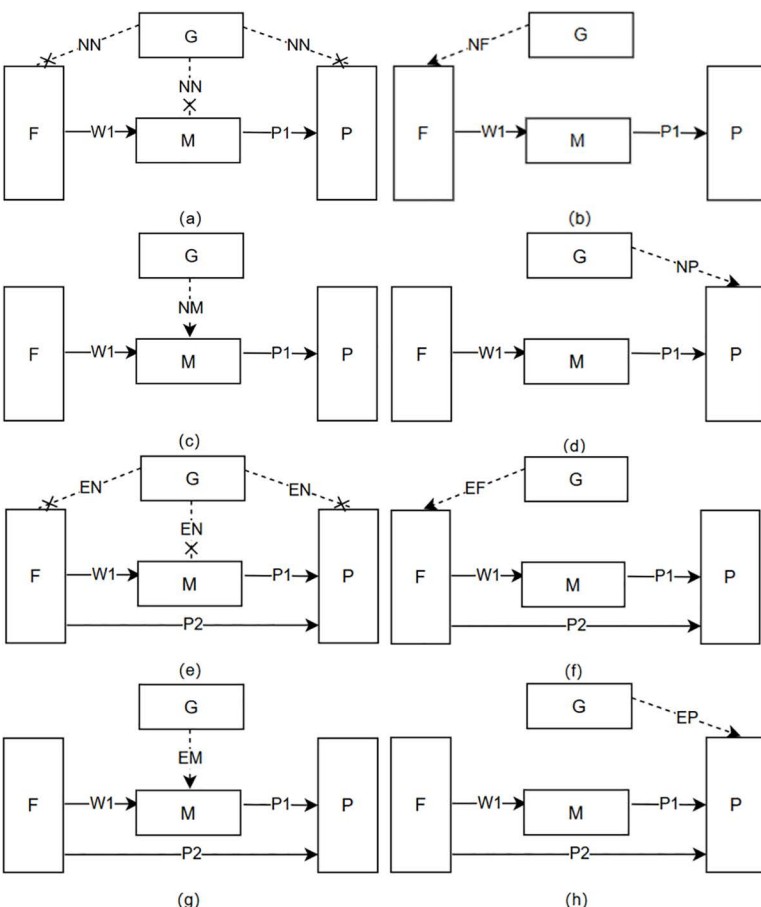

**Fig 1. Eight scenarios in which the government subsidizes members of the biomass power supply chain.**

The social welfare (the combined benefits of all stakeholders in the biomass power generation supply chain, including both producer profits and consumer surplus [51]) formula is:

$$SW = CS + \pi_F + \pi_M + \pi_{P.} \tag{5}$$

To enhance social welfare and improve the stability of the biomass supply chain, the government may provide subsidies to relevant participants within the biomass supply chain. Typically, subsidies are allocated to farmers, middlemen, or biomass power plants. For example, rice straw collection and baling in Fenghua District, Ningbo City, in 2023 was subsidized at a rate of RMB 75/ton [52]. Similarly, unit subsidies were provided to middlemen and biomass power plants in Hepu County, Guangxi Province, where Hepu County Guanxing Modern Agricultural Development Co. Ltd., acting as a middleman, received a subsidy of RMB 40/ton for recycling and utilizing 15,041.790 tons of crop straw, while Hepu Liang Agricultural and Forestry Waste Cogeneration Co., Ltd. was subsidized at RMB 50/ton for processing 3253.6289 tons of crop straw [53]. For simplicity, it is assumed that the government provides a subsidy of m (m > 0) per unit to biomass supply chain members [35].

As illustrated in Fig 2, the sequence of decision-making follows these steps: (1) Farmers determine whether to engage in encroachment; (2) The government selects a subsidy policy. To assess the impact of decision order on equilibrium outcomes, an extension is considered where the government's subsidy decision precedes the farmer's encroachment strategy, confirming the robustness of the main results; (3) The middleman sets the unit price for selling biomass to the power plant $p_1$; (4) Farmers determine the wholesale price of biomass sold to middlemen $w_1$ and the unit price of biomass sold directly to power plants $p_2$ The sequence of decisions between farmers and middlemen is detailed in the corresponding table. This decision-making framework highlights the intricate interactions and strategic competition among farmers, middlemen, and power plants. Essentially, the biomass power supply chain operates as a supply system where electricity production serves as the primary driving force [54]. Additionally, this section clarifies the price decision-making process corresponding to stages 3 and 4 [55].

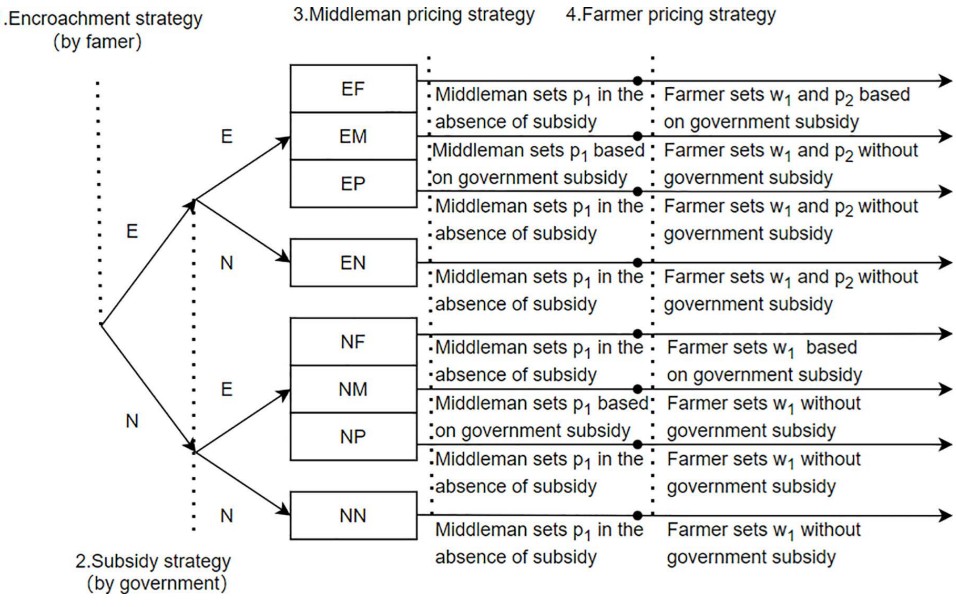

**Fig 2. Sequence of events.**

Next, the equilibrium wholesale price and equilibrium unit selling price for each of the eight subgames will be derived. The equilibrium profits of supply chain members and the resulting social welfare under different strategic scenarios will be calculated. Finally, the interaction between farmers' channel encroachment and government subsidy strategies will be analyzed.

## 4. Government subsidy scenarios

This chapter examines different scenarios in which government subsidies influence members of the biomass supply chain, deriving the pricing equilibrium strategies for each specific case.

### 4.1. Scenario NN

In scenario NN, farmers supply biomass feedstock to middlemen, who subsequently sell it to power plants, with no policy subsidies provided to any biomass supply chain members by the relevant authorities. Therefore, the profit function for each supply chain member in this scenario is given by:

$$\pi_P = d_1^N (p_0 t - c_3), \tag{6}$$

$$\pi_M = d_1^N (p_1 - w_1 - c_2), \tag{7}$$

$$\pi_F = d_1^N (w_1 - c_1) \tag{8}$$

Using the inverse solution method, first, let (7) be the first-order derivative of $p_1$ and take the first-order derivative to obtain the reaction function $p_1 = \frac{1}{2}(a + c_2 + w_1)$, which yields a reaction function. Bringing this reaction function into (8) and then letting (8) take the first-order derivative with respect to $w_1$ find the first-order derivative, we can obtain $w_1 = \frac{1}{2}(a + c_1 - c_2)$. Bringing $w_1 = \frac{1}{2}(a + c_1 - c_2)$ into the above $p_1 = \frac{1}{2}(a + c_2 + w_1)$ yields the equilibrium price.

### 4.2. Scenario NF

In Scenario NF, farmers supply biomass feedstock to middlemen, who subsequently sell it to power plants, and the relevant authorities provide policy subsidies to biomass supply chain member farmers. Therefore, the profit function for each supply chain member in this scenario is given by:

$$\pi_P = d_1^N (p_0 t - c_3), \tag{9}$$

$$\pi_M = d_1^N (p_1 - w_1 - c_2), \tag{10}$$

$$\pi_F = d_1^N (w_1 - c_1 + m) \tag{11}$$

Using the inverse solution method: first let (10) be the first order derivative of $p_1$ and take the first order derivative to obtain the reaction function $p_1 = \frac{1}{2}(a + c_2 + w_1)$. Bringing this reaction function into (11) and then letting (11) take the first order derivative with respect to $w_1$ find the first order derivative, we can obtain $w_1 = \frac{1}{2}(a + c_1 - c_2 + m)$. Bringing $w_1 = \frac{1}{2}(a + c_1 - c_2 + m)$ into the above $p_1 = \frac{1}{2}(a + c_2 + w_1)$ yields the equilibrium price.

### 4.3. Scenario NM

In scenario NM, farmers supply biomass feedstock to middlemen, who subsequently sell it to power plants, and the relevant authorities provide policy subsidies to biomass supply chain member middlemen. Therefore, the profit function for each supply chain member in this scenario is given by:

$$\pi_P = d_1^N (p_0 t - c_3),$$
(12)

$$\pi_M = d_1^N (p_1 - w_1 - c_2 + m),$$
(13)

$$\pi_F = d_1^N (w_1 - c_1)$$
(14)

Using the inverse solution method: first let (13) be the first order derivative of $p_1$ and take the first order derivative to obtain the reaction function $p_1 = \frac{1}{2}(a + c_2 - m + w_1)$, which yields a reaction function. Bringing this reaction function into (14) and then letting (14) take the first order derivative with respect to $w_1$ find the first-order derivative, we can obtain $w_1 = \frac{1}{2}(a + c_1 - c_2 + m)$. Bringing $w_1 = \frac{1}{2}(a + c_1 - c_2 + m)$ into the above $p_1 = \frac{1}{2}(a + c_2 - m + w_1)$ yields the equilibrium price.

### 4.4. Scenario NP

In scenario NP, farmers supply biomass feedstock to middlemen, who subsequently sell it to power plants, and the relevant authorities provide policy subsidies to biomass supply chain member power plants. Therefore, the profit function for each supply chain member in this scenario is given by:

$$\pi_P = d_1^N (p_0 t - c_3 + m),$$
(15)

$$\pi_M = d_1^N (p_1 - w_1 - c_2),$$
(16)

$$\pi_F = d_1^N (w_1 - c_1)$$
(17)

Using the inverse solution method: first let (16) be the first order derivative of $p_1$ and take the first order derivative to obtain the reaction function $p_1 = \frac{1}{2}(a + c_2 + w_1)$ to obtain the reaction function Bringing this reaction function into (17) and then letting (17) take the first order derivative with respect to $w_1$ find the first-order derivative, we can obtain $w_1 = \frac{1}{2}(a + c_1 - c_2)$. Bringing $w_1 = \frac{1}{2}(a + c_1 - c_2)$ into the above $p_1 = \frac{1}{2}(a + c_2 + w_1)$ yields the equilibrium price. For convenience, we put the equilibrium pricing results in Table 3.

### 4.5. Scenario EN

In Scenario EN, farmers supply biomass feedstock directly to power plants in addition to selling it through middlemen, and no policy subsidies are granted to biomass supply chain members by the relevant authorities. Therefore, the profit function of each supply chain member in this scenario is given by:

$$\pi_P = (p_0 t - c_3)(d_1^E + d_2)$$
(18)

$$\pi_M = d_1^E (p_1 - w_1 - c_2)$$
(19)

$$\pi_F = d_1^E (w_1 - c_1) + d_2 (p_2 - c_0)$$
(20)

**Table 3. Equilibrium pricing in the no-encroachment channel scenarios.**

| Scenario | $w_1$ | $p_1$ |
|---|---|---|
| NN | $\frac{1}{2}(a + c_1 - c_2)$ | $\frac{1}{4}(3a + c_1 + c_2)$ |
| NF | $\frac{1}{2}(a + c_1 - c_2 - m)$ | $\frac{1}{4}(3a + c_1 + c_2 - m)$ |
| NM | $\frac{1}{2}(a + c_1 - c_2 + m)$ | $\frac{1}{4}(3a + c_1 + c_2 - m)$ |
| NP | $\frac{1}{2}(a + c_1 - c_2)$ | $\frac{1}{4}(3a + c_1 + c_2)$ |

Using the inverse solution method, first, let (19) be a first-order derivative of $p_1$ taking the first-order derivative, which gives $p_1 = \frac{1}{2}(a + c_2 + bp_2 + w_1)$ Bringing this reaction function into (20), let the profit function (20) take a first-order derivative with respect to $w_1$ and $p_2$ find the first-order derivatives, respectively, to obtain the reaction function $w_1 = \frac{-a-c_1 + bc_1 + c_2 - bc_2}{2(-1 + b)}$, $p_2 = \frac{-a-c_0 + bc_0}{2(-1 + b)}$ and the first-order derivative of the profit function (20) with respect to Bringing the $w_1 = \frac{-a-c_1 + bc_1 + c_2 - bc_2}{2(-1 + b)}$, $p_2 = \frac{-a-c_0 + bc_0}{2(-1 + b)}$ into the above $p_1 = \frac{1}{2}(a + c_2 + bp_2 + w_1)$ to obtain the equilibrium result.

### 4.6. Scenario EF

In Scenario EF, farmers supply feedstock directly to the power plant in addition to selling it through middlemen, and the relevant authorities provide policy subsidies to biomass supply chain member farmers. Therefore, the profit function of each supply chain member in this scenario is given by:

$$\pi_P = (p_0 t - c_3)\left(d_1^E + d_2\right), \tag{21}$$

$$\pi_M = d_1^E (p_1 - w_1 - c_2), \tag{22}$$

$$\pi_F = d_1^E (w_1 - c_1 + m) + d_2 (p_2 - c_0 + m), \tag{23}$$

Using the inverse solution method: first let (22) be a first order derivative of $p_1$ taking the first-order derivative, which gives $p_1 = \frac{1}{2}(a + c_2 + bp_2 + w_1)$ Bringing this reaction function into (23), let the profit function (23) take the first order derivative with respect to $w_1$ and $p_2$ find the first-order derivatives, respectively, to obtain the reaction function $w_1 = \frac{-a-c_1 + bc_1 + c_2 - bc_2 + m - bm}{2(-1 + b)}$, $p_2 = \frac{-a-c_0 + bc_0 + m - bm}{2(-1 + b)}$ and the first order derivative of the profit function (23). Bringing the $w_1 = \frac{-a-c_1 + bc_1 + c_2 - bc_2 + m - bm}{2(-1 + b)}$, $p_2 = \frac{-a-c_0 + bc_0 + m - bm}{2(-1 + b)}$ into the above $p_1 = \frac{1}{2}(a + c_2 + bp_2 + w_1)$ to obtain the equilibrium result.

### 4.7. Scenario EM

In Scenario EM, farmers supply feedstock directly to power plants in addition to selling it through middlemen, and the relevant authorities provide policy subsidies to biomass supply chain member middlemen. Therefore, the profit function of each supply chain member in this scenario is given by:

$$\pi_P = (p_0 t - c_3)\left(d_1^E + d_2\right), \tag{24}$$

$$\pi_M = d_1^E (p_1 - w_1 - c_2 + m), \tag{25}$$

$$\pi_F = d_1^E (w_1 - c_1) + d_2 (p_2 - c_0),$$ (26)

Using the inverse solution method: first let (25) be a first-order derivative of $p_1$ first-order derivatives, we get $p_1 = \frac{1}{2}(a + c_2 - m + bp_2 + w_1)$ Bringing this reaction function into (26), let the profit function (26) take the first-order derivative with respect to $w_1$ and $p_2$ and the first-order derivatives of (26), respectively, to obtain the reaction function $w_1 = \frac{-a - c_1 + bc_1 + c_2 - bc_2 - m + bm}{2(-1 + b)}, p_2 = \frac{-a - c_0 + bc_0}{2(-1 + b)}$. Bringing $w_1 = \frac{-a - c_1 + bc_1 + c_2 - bc_2 - m + bm}{2(-1 + b)}, p_2 = \frac{-a - c_0 + bc_0}{2(-1+b)}$ into the above $p_1 = \frac{1}{2}(a + c_2 - m + bp_2 + w_1)$ to obtain the equilibrium result.

### 4.8. Scenario EP

In Scenario EP, farmers supply feedstock directly to power plants in addition to selling it through middlemen, and the relevant authorities provide policy subsidies to biomass supply chain member power plants. Therefore, the profit function of each supply chain member in this scenario is given by:

$$\pi_P = (p_0 t - c_3 + m)(d_1^E + d_2),$$ (27)

$$\pi_M = d_1^E (p_1 - w_1 - c_2),$$ (28)

$$\pi_F = d_1^E (w_1 - c_1) + d_2 (p_2 - c_0)$$ (29)

Using the inverse solution method: first let (28) take the first-order derivative of $p_1$ taking the first order derivative, which gives $p_1 = \frac{1}{2}(a + c_2 + bp_2 + w_1)$ Bringing this reaction function into (29), let the profit function (29) take a first-order derivative with respect to $w_1$ and $p_2$ find the first-order derivatives, respectively, to obtain the reaction function $w_1 = \frac{-a - c_1 + bc_1 + c_2 - bc_2}{2(-1 + b)}, p_2 = \frac{-a - c_0 + bc_0}{2(-1 + b)}$ and the first-order derivative of the profit function (29). Bringing the $w_1 = \frac{-a - c_1 + bc_1 + c_2 - bc_2}{2(-1 + b)}, p_2 = \frac{-a - c_0 + bc_0}{2(-1 + b)}$ into the above $p_1 = \frac{1}{2}(a + c_2 + bp_2 + w_1)$ to obtain the equilibrium result.

The equilibrium results for the four scenarios under channel encroachment are summarized in Table 4.

## 5. Analysis of balanced results

**Proposition 1:** For $w_1$ and $p_1$ for the farmer-no-encroachment case: $w_1^{NF} < w_1^{NM} < w_1^{NN} = w_1^{NP}$, the $p_1^{NF} = p_1^{NM} < p_1^{NN} = p_1^{NP}$, the encroachment case $w_1^{EF} < w_1^{EN} = w_1^{EP} < w_1^{EM}.p_1^{EF} < p_1^{EM} < p_1^{EP} = p_1^{EN}$ $.p_2^{EF} < p_2^{EN} = p_2^{EM} = p_2^{EP}$.

Proposition 1 illustrates the impact of government subsidy strategies and farmer channel encroachment on equilibrium prices. When farmers do not engage in encroachment, subsidies provided to upstream farmers or middlemen enhance their marginal profits. As a result, farmers or middlemen lower prices and increase sales to maximize profits. Consequently, in a single-channel scenario, subsidizing either upstream farmers or middlemen will make the $w_1$ and $p_1$ decrease. In contrast, in the case of a farmer's choice of channel encroachment, a government subsidy to a middleman may make $w_1$ up. This occurs because subsidies directed solely at middlemen enable them to attain higher marginal profits, prompting farmers to raise the wholesale price to secure better returns. Finally, it is evident that regardless of encroachment, when government subsidies are granted to power plants, wholesale and retail prices remain unchanged, as power plants do not participate in price-setting decisions.

**Proposition 2:** For farmers, when there is no encroachment, $\pi_F^{NP} = \pi_F^{NN} < \pi_F^{NM} = \pi_F^{NF}$; when there is encroachment, $\pi_F^{EN} < \pi_F^{EM} < \pi_F^{EF} = \pi_F^{EP}$. To highlight the conclusions, we assigned values to the parameters and compared the impact of government subsidies on the equilibrium income of farmers. As shown in Fig 3, let $c_1 = 2, c_2 = 1, c_3 = 0.8, c_0 = 0.3, a = 5, b = 0.6$.

 

**Table 4. Equilibrium pricing in the encroachment channel scenarios.**

| Scenario | $w_1$ | $p_1$ | $p_2$ |
|---|---|---|---|
| EN | $\dfrac{-a-c_1+bc_1+c_2-bc_2}{2(-1+b)}$ | $\dfrac{-3a+ab-bc_0+b^2c_0-c_1+bc_1-c_2+bc_2}{4(-1+b)}$ | $\dfrac{-a-c_0+bc_0}{2(-1+b)}$ |
| EF | $\dfrac{-a-c_1+bc_1+c_2-bc_2+m-bm}{2(-1+b)}$ | $\dfrac{-3a+ab-bc_0+b^2c_0-c_1+bc_1-c_2+bc_2+m-b^2m}{4(-1+b)}$ | $\dfrac{-a-c_0+bc_0+m-bm}{2(-1+b)}$ |
| EM | $\dfrac{-a-c_1+bc_1+c_2-bc_2-m+bm}{2(-1+b)}$ | $\dfrac{-3a+ab-bc_0+b^2c_0-c_1+bc_1-c_2+bc_2+m-bm}{4(-1+b)}$ | $\dfrac{-a-c_0+bc_0}{2(-1+b)}$ |
| EP | $\dfrac{-a-c_1+bc_1+c_2-bc_2}{2(-1+b)}$ | $\dfrac{-3a+ab-bc_0+b^2c_0-c_1+bc_1-c_2+bc_2}{4(-1+b)}$ | $\dfrac{-a-c_0+bc_0}{2(-1+b)}$ |

Proposition 2 illustrates the impact of government subsidies on farmers' profits under different encroachment scenarios. The findings indicate that in the absence of encroachment, subsidies provided to either middlemen or farmers lead to increased profits for farmers [16], whereas subsidies directed toward power plants do not enhance farmers' profits. When subsidies are allocated to farmers, their production costs decrease, affording them greater pricing flexibility. Subsidies to middlemen enable them to allocate more resources toward market expansion and operational improvements, prompting them to raise purchase prices to secure a stable supply of raw materials, thereby increasing farmers' profits. Since power plants do not engage directly with farmers in pricing transactions, subsidies granted to power plants do not significantly affect farmers' profits. In cases where farmers engage in encroachment, subsidies provided directly to farmers yield the highest profits, followed by subsidies to middlemen, and lastly, subsidies to power plants. When farmers receive subsidies, their marginal profits increase, incentivizing them to expand supply through all available channels to maximize earnings. If middlemen receive subsidies, their purchasing capacity increases, leading to greater coordination flexibility in pricing $w_1$. This results in more opportunities for farmers to benefit from middlemen channels.

**Proposition 3:** For the middleman, when there is no encroachment, $\pi_M^{NP} = \pi_M^{NN} < \pi_M^{NM} = \pi_M^{NF}$; when there is an invasion, $\pi_M^{EN} = \pi_M^{EP} < \pi_M^{EF} < \pi_M^{EM}$. To highlight the conclusions, we assigned values to the parameters

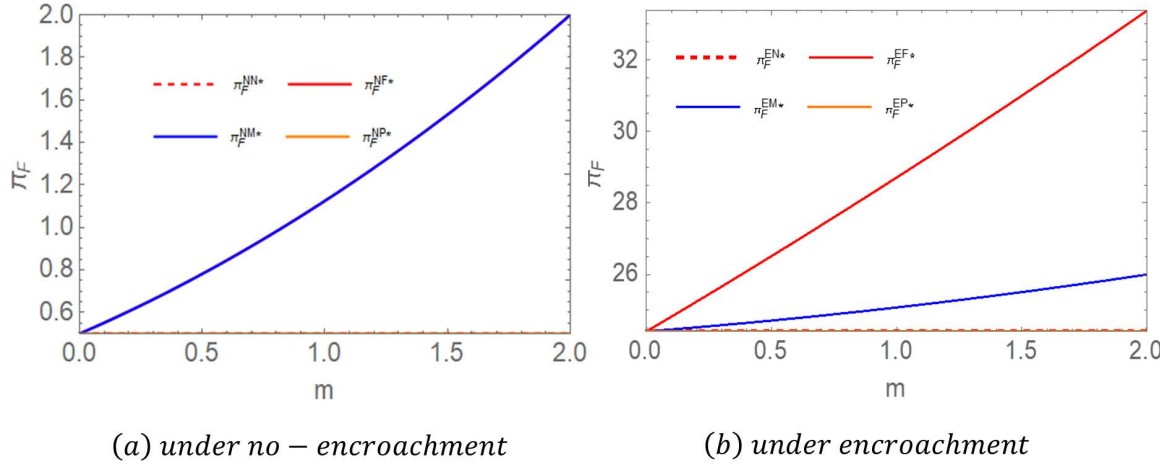

(a) under no − encroachment          (b) under encroachment

**Fig 3. Comparative chart of farmers' equilibrium returns.**

and compared the impact of government subsidies on the equilibrium income of middlemen. As shown in Fig 4, let $c_1 = 2, c_2 = 1, c_3 = 0.8, c_0 = 0.3, a = 5, b = 0.6$.

Proposition 3 illustrates the impact of government subsidies on middlemen's profits under different encroachment scenarios. The results in the no-encroachment scenario align with those of Proposition 2, requiring little additional explanation. In the encroachment scenario, subsidies provided to middlemen yield the highest profits, followed by subsidies to farmers, and finally, subsidies to power plants. When middlemen receive subsidies, their marginal profits increase, offsetting the losses incurred due to farmers' encroachment into their supply channel. When farmers are subsidized, their willingness to supply biomass increases. To achieve economies of scale [56], they tend to lower prices to attract larger orders. These price reductions significantly decrease the procurement costs for middlemen while simultaneously providing farmers with stable sales volumes and incomes.

**Proposition 4:** For biomass power plants, when there is no encroachment, $\pi_P^{NN} < \pi_P^{NM} = \pi_P^{NF} < \pi_P^{NP}$; when there is invasion, $\pi_P^{EN} < \pi_P^{EM} < \pi_P^{EF} < \pi_P^{EP}$. To show the conclusions more clearly, we compare the equilibrium returns of biomass power plants under different scenarios. As shown in Fig 5, let $c_1 = 1.2, c_2 = 0.8, c_3 = 0.5, c_0 = 0.3, a = 5, b = 0.6, p_0 = 2.5$.

Proposition 4 illustrates the impact of government subsidies on biomass power plant profits under different encroachment scenarios. The results indicate that providing a unit subsidy to biomass power plants maximizes their profits regardless of whether encroachment occurs. When a power plant receives a subsidy, it offsets the significant costs associated with biomass feedstock procurement [57]. In the absence of channel encroachment, subsidies to either farmers or middlemen improve power plant profitability. However, in the presence of encroachment, subsidizing farmers proves to be more effective than subsidizing middlemen. Subsidizing farmers strengthens cooperative relationships within the supply chain, fostering a more stable and long-term feedstock supply arrangement with middlemen or power plants. This stability, whether through direct (encroachment) or indirect (no encroachment) means, supports the continuous operation and profit growth of biomass power plants. Conversely, when subsidies are directed toward middlemen in an encroachment scenario, middlemen gain greater bargaining power, which can lead to a $p_2$ consequent rise, then squeezes biomass power plant profits.

**Proposition 5:** In the no-encroachment scenario, subsidies to middlemen or farmers maximize social welfare; in the encroachment scenario, subsidies to farmers alone maximize social welfare. To visualize the results more, m is marked as the amount of subsidy per unit of biomass material; SW is the total social welfare; and $c_1 = 1.25, c_2 = 1, c_3 = 0.8, c_0 = 0.3, a = 3, p_0 = 2.5$, a government decision diagram, as shown in Fig 6, can be obtained:

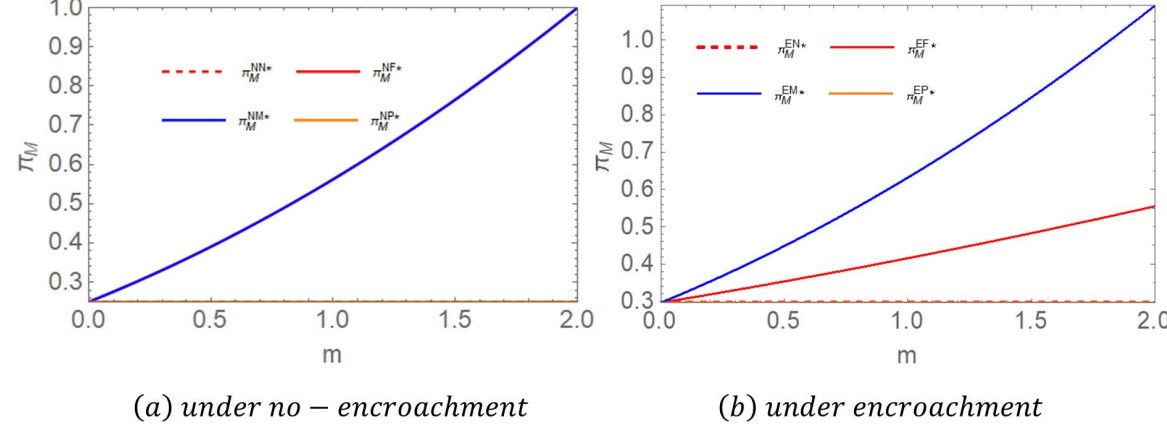

(a) under no − encroachment        (b) under encroachment

**Fig 4. Comparative chart of middlemen's equilibrium returns.**

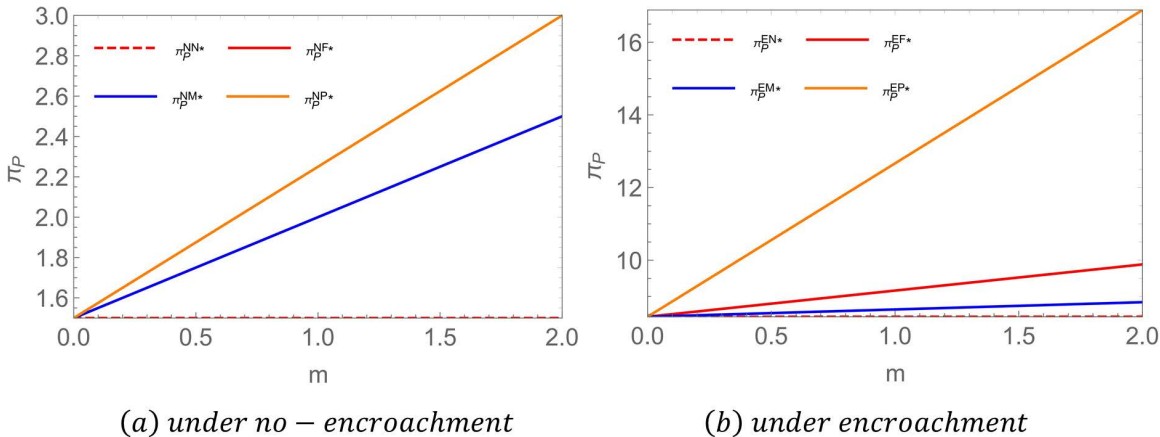

$(a)$ *under no − encroachment*    $(b)$ *under encroachment*

**Fig 5. Comparison of equilibrium returns from biomass power plants.**

Proposition 5 demonstrates that in the absence of encroachment, government subsidies consistently enhance social welfare. This is supported by the findings of Proposition 1, which indicate that subsidizing either middlemen or farmers leads to reductions in wholesale and retail prices. Consequently, the profits of farmers and middlemen increase, ultimately contributing to higher overall profits within the biomass power supply chain. Additionally, the increase in sales volume raises consumer surplus. The simultaneous growth in consumer surplus and supply chain profits results in an overall improvement in social welfare. In the presence of encroachment, targeted subsidies for farmers stimulate biomass supply efficiency from the production stage. Under this time $w_1^{EF}$, $P_1^{EF}$, $P_2^{EF}$ (wholesale price of biomass straw and the selling price per unit in each channel) are minimized, leading to the maximization of social welfare.

**Proposition 6:** In the encroachment scenario, the probability of harming social welfare in the EM case is smaller when the intensity of channel competition is smaller. To visualize the result more, let $c_1 = 1.25, c_2 = 1, c_3 = 0.8, c_0 = 0.3, a = 3, p_0 = 2.5$, where the left pane $b = 0.4$, the right diagram $b = 0.8$, The resulting government decision diagram is presented in Fig 7.

Proposition 6 suggests that, unlike in the no-encroachment scenario, government subsidies directed solely to middlemen in the encroachment scenario may be detrimental to social welfare, particularly when subsidy intensity is low. This occurs because, in the presence of government subsidies for middlemen, competition between the existing supply channel and the encroachment channel intensifies. When subsidy intensity is insufficient, the increase in subsidized profits is not substantial enough to offset profit losses caused by heightened competition, ultimately leading to a decline in overall social welfare. Furthermore, the intensity of channel competition significantly influences this trend—greater competition increases the likelihood of an overall reduction in social welfare.

**Proposition 7:** The equilibrium outcome of the entire game is determined as EF by solving the game in reverse, indicating that farmers will opt to establish additional marketing channels, while the government will provide subsidies specifically for farmers.

Proposition 7 suggests that farmers will develop direct supply channels, with government subsidies supporting their expansion. Regardless of whether farmers choose to encroach, subsidies remain the most effective tool for incentivizing farmers to supply biomass raw materials and ensure the sustainable development of the biomass power generation supply chain. In particular, when farmers do not engage in encroachment, the government may subsidize either farmers or middlemen. However, when encroachment occurs, subsidies are directed exclusively to farmers. Although encroachment reduces the revenues of middlemen, the increased profitability of the encroachment channel compensates for these losses, making the supply chain as a whole more profitable. Consequently, in pursuit of higher profits, farmers are

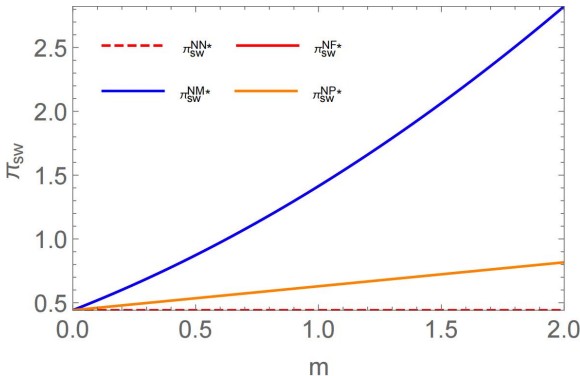

**Fig 6. Impact of different subsidy recipients on social welfare in the no-encroachment scenario.**

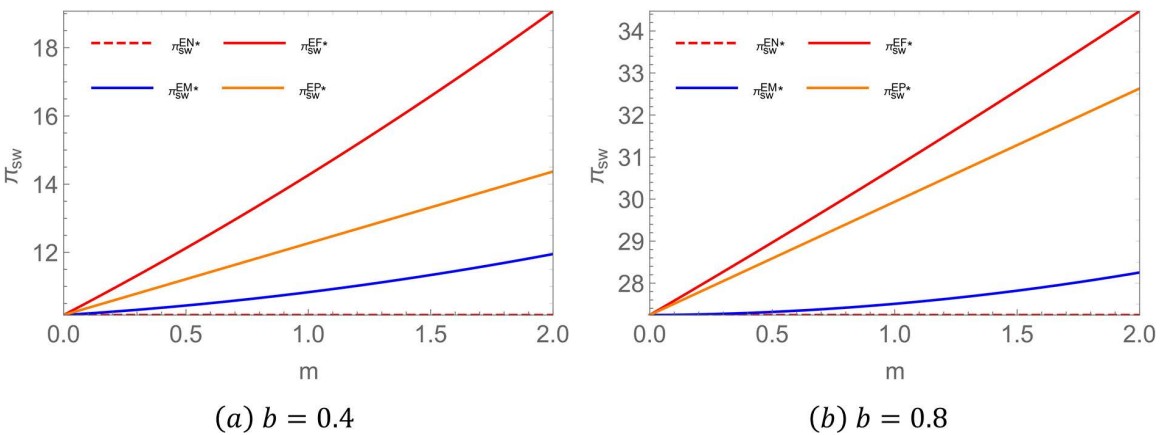

$(a)\ b = 0.4$ $(b)\ b = 0.8$

**Fig 7. Impact of different subsidy recipients on social welfare in the encroachment scenario.**

incentivized to expand their sales channels strategically, inducing the government to allocate subsidies solely to farmers, thereby securing maximum returns.

## 6. Conclusion

This study analyzes government subsidy strategies for biomass power supply chain members across different supply channels and examines the interactions between these strategies and the involved stakeholders. A biomass power generation supply chain system model was developed, incorporating three key participants: farmers *(F)*, intermediaries *(M)*, and power generation enterprises *(P)*. Within this multi-tiered supply chain structure, agricultural producers are responsible for collecting and initially processing biomass raw materials, supplying them to intermediaries through a wholesale trade mechanism. Intermediaries, in turn, transfer the processed raw materials to power generation enterprises for energy conversion. Notably, the system accommodates vertical integration, enabling agricultural producers to bypass intermediaries and establish direct supply channels with power generation enterprises $(F \rightarrow P)$ under specific market conditions, forming a hybrid supply chain model. As the policy-regulating entity, the government is responsible for optimizing the selection of subsidy recipients $(F, M, P)$, determining the appropriate subsidy intensity, and designing effective subsidy mechanisms. These interventions aim to achieve multiple policy objectives, including maximizing energy system efficiency and

enhancing social welfare. This study employs game theory to evaluate the potential impact of government subsidy mechanisms on supply chain performance. The findings contribute to a deeper understanding of the crucial role subsidy policies play in improving the efficiency and sustainability of biomass supply chain management.

## 6.1. Main conclusions

This study examined eight strategic scenarios within the biomass power generation supply chain, leading to the following key conclusions. First, equilibrium prices for supply chain participants are significantly affected by government subsidy strategies and channel encroachment behaviors. Specifically, regardless of whether encroachment occurs, subsidies directed toward farmers or intermediaries reduce the raw material procurement cost for biomass power plants, with subsidies to farmers yielding the lowest supply price for biomass raw materials [58]. Second, the analysis demonstrates that in the absence of encroachment, subsidizing upstream supply chain members, such as intermediaries and farmers, maximizes social profits. However, contrary to existing theories [59], the findings indicate that under encroachment conditions, inadequate government subsidy intensity may negatively impact social profits. Therefore, government agencies must strategically determine subsidy recipients and levels based on the degree of channel competition. This study provides a new perspective on government incentive strategies for biomass power supply chains and offers practical guidance for policymakers responsible for subsidy allocation. Finally, the research findings suggest that farmers benefit from establishing direct sales channels, while government subsidies play a crucial role in supporting these transitions. This strategy mitigates the loss of intermediary profits by increasing farmers' earnings through higher direct sales prices. For example, in Bozhou, Anhui Province, the straw purchase price from middlemen is about 240 yuan per ton, whereas direct sales to power plants yield around 290 yuan per ton [60]. Additionally, government subsidies enhance farmers' willingness to supply biomass [12]. In Ketian Town, Yanhe County, Guizhou Province, a subsidy of 200 yuan per ton for straw has resulted in an expected recovery of 1,900 tons of straw in 2024, generating an estimated total output value of 600,000 yuan [61].

## 6.2. Management Insights

This study provides insights into the government's subsidy decisions within the biomass power supply chain. Unlike conventional perspectives, this research examines how government subsidy policies can be designed to maximize social welfare under different scenarios, particularly when biomass feedstock suppliers (farmers) either engage in channel encroachment or continue traditional supply arrangements. These findings have important implications for the regulation and sustainable development of the biomass power generation supply chain.

Two key factors influencing the government's optimal subsidy strategy are identified: channel structure and channel competition intensity. These findings suggest that government agencies should dynamically adjust their subsidy policies based on different market conditions. (1) Whether supply channel competition exists or not, government subsidies for farmers are crucial for ensuring the stable supply of biomass raw materials. In practice, farmers often exhibit low willingness to supply straw and frequently resort to field burning, leading to resource wastage [62]. The government can leverage subsidy policies to address this issue. Firstly, it can encourage biomass power generation enterprises to establish long-term and stable cooperative relationships with farmers by providing additional subsidies for joint projects. This promotes information sharing and collaborative operations between upstream and downstream supply chain partners, thereby enhancing overall supply chain efficiency. Secondly, policy preferences and financial support can be offered to farmers who actively participate in building the coordination mechanism of the biomass power generation supply chain. This fosters a favorable market environment and enhances supply chain stability. (2) Interestingly, our research demonstrates that the government should exercise caution when providing subsidies to intermediaries. In the absence of external competition, subsidies can effectively lower operational costs for intermediaries, thereby ensuring they remain motivated to organize and coordinate the supply of biomass raw materials, thus enhancing supply stability. However, in the presence of external competition, the government must carefully consider the level of subsidies

provided to intermediaries. Poorly designed subsidies may exacerbate channel competition. (3) When the intensity of channel competition is high, an excessively low subsidy coefficient fails to adequately incentivize key links in the biomass power generation supply chain. This leads to unstable raw material supply, which negatively impacts the production efficiency of biomass power generation enterprises and ultimately reduces social welfare. Additionally, an overly low subsidy coefficient may result in cost transfers across the supply chain, causing increases in the price of biomass power generation products and reductions in consumer surplus, further diminishing social welfare. Particularly, when subsidies are provided to intermediaries under such conditions, it may exacerbate unnecessary price competition. It is recommended that the government, considering long-term perspectives and incorporating the findings of this study, develop forward-looking subsidy policies.

### 6.3. Future research

This study has certain limitations. First, it assumes that the government provides a per-unit subsidy to participants in the biomass supply chain. While unit price subsidies are prevalent in biomass power supply chains, subsidies in practice also take other forms, such as linear growth subsidies [63] and fixed rebate subsidies [64]. Future research could explore a combination of different subsidy models to provide a more comprehensive understanding of their effects. Second, government subsidies to biomass supply chain members involve specific subsidy coefficients and risk factors [8]. The uncertainty associated with subsidy costs and potential risks may influence government subsidy strategies. Further investigation into the impact of subsidy risks and dynamic subsidy costs on government decision-making would be valuable. Finally, this study focuses on competition between intermediaries and farmers within the biomass supply chain. However, in real-world scenarios, horizontal competition among intermediaries for biomass materials also plays a critical role [65]. Future research should examine this aspect to gain deeper insights into its implications for supply chain efficiency and policy optimization.

## Supporting information

**S1 File.  Appendix 1 Derivation process.**
(DOCX)

## Author contributions

**Conceptualization:** Xin Wu, Guangyin Xu.

**Data curation:** Xin Wu, Peng Liu.

**Formal analysis:** Xin Wu.

**Funding acquisition:** Heng Wang.

**Investigation:** Xin Wu, Peng Liu, Jin Li, Jing Gao, Hang Ke.

**Methodology:** Xin Wu, Jin Li, Jing Gao, Guangyin Xu, Hang Ke.

**Project administration:** Heng Wang.

**Software:** Guangyin Xu.

**Validation:** Xin Wu.

**Visualization:** Xin Wu, Peng Liu.

**Writing – original draft:** Xin Wu.

**Writing – review & editing:** Xin Wu, Peng Liu, Guangyin Xu.

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
