## [Decision Letter · Decision Letter 0]

24 Jan 2025

PONE-D-24-49884Research on government subsidy strategy of biomass power supply chain considering channel encroachmentPLOS ONE

Dear Dr. Xu,

Thank you for submitting your manuscript to PLOS ONE. After careful consideration, we feel that it has merit but does not fully meet PLOS ONE’s publication criteria as it currently stands. Therefore, we invite you to submit a revised version of the manuscript that addresses the points raised during the review process.

I recommend that it should be revised taking into account the changes requested by the reviewers. Since the requested changes include valuable and constructive reviews, I would like to give you a chance to revise your manuscript. The revised manuscript will undergo the next round of review by same reviewers.

We look forward to receiving your revised manuscript.

Kind regards,

Baogui Xin, Ph.D.

Academic Editor

PLOS ONE

Journal Requirements:

 “Project Title: "Route Planning for Fresh Agricultural Product Delivery Vehicles under Traffic Control Policies of Restricted Access," Henan Province Scientific and Technological Research Project, 2024, No. 242102240028.” 

3. We note that your Data Availability Statement is currently as follows: “All relevant data are within the manuscript and in Supporting Information files.”

4. Please ensure that you refer to Figure 1 and 2 in your text as, if accepted, production will need this reference to link the reader to the figure.

Reviewers' comments:

Reviewer's Responses to Questions

**Comments to the Author**

1. Is the manuscript technically sound, and do the data support the conclusions?

Reviewer #1: Yes

Reviewer #2: Yes

Reviewer #3: Yes

2. Has the statistical analysis been performed appropriately and rigorously? 

Reviewer #1: No

Reviewer #2: Yes

Reviewer #3: Yes

3. Have the authors made all data underlying the findings in their manuscript fully available?

Reviewer #1: Yes

Reviewer #2: Yes

Reviewer #3: Yes

4. Is the manuscript presented in an intelligible fashion and written in standard English?

Reviewer #1: No

Reviewer #2: No

Reviewer #3: Yes

5. Review Comments to the Author

Reviewer #1: This study presents the strategic interaction between government subsidy strategies and farmers' channel encroachment strategies in a biomass power supply chain. This is an interesting topic. However, some issues are provided below.

(1) Part I, Introduction and Related work seems to be logically confusing. The author should clearly indicate the main idea of each paragraph.

(2) Many Tables go beyond the margins.

(3) The images in the text are not clear enough. They are blurry, making it difficult to see the numbers and symbols.

(4) The format of the references is also messy.

(5) The English expression in the article needs improvement.

Reviewer #2: This manuscript investigates government subsidy strategies within the biomass power supply chain under different scenarios of channel competition and encroachment. Using a game-theoretic model, the authors analyze eight subsidy scenarios and identify conditions that optimize profits and social welfare. Key findings include that subsidies to farmers consistently maximize social welfare, while subsidies to middlemen may harm welfare in high-competition scenarios. The equilibrium suggests that farmers prefer encroachment to induce subsidies.

The discussion in this paper is complete. If it can be improved in quality, it can be considered for publication. I have the following questions and points:

- It is better to introduce the definition of “standard Stackelberg game setup”. Since there are games, are there strategies? Are strategies fixed or evolutionary?

- I encourage the authors to discuss why is it necessary to divide eight scenarios. How does this way of dividing between scenarios contribute to the core conclusions of this paper?

- The format of citations is not consistent.

- “Scenario NN (Fig a)” and similar later on: which figure?

- An optional suggestion: change “There is no channel competition, and the relevant sector does not subsidize the supply chain members” to “There is No channel competition, and the relevant sector does Not subsidize the supply chain members” and similar later on, would make it easier for readers.

- “similar to Jiang et al. 's study”. You missed a citation.

- Please write more in figure captions, making a figure self-contained.

- Punctuate each independent formula to ensure sentences have an end.

- There are many non-English punctuation marks.

- I did not see the definitions of CS and SW.

- Phrases like “the Wu[23] article” are weird.

- “this chapter”. There is no chapter in a journal article, only sections.

- Table 2: I did not see the definitions of W_1 and P_1. What is the difference between W and w? What is the difference between P and p?

- Font sizes in figures are unacceptable.

- English proofreading and polishing are needed. All kinds of formats need to be error-free. Any error will influence the acceptance of the manuscript.

Reviewer #3: 0.The abstract is dense, making it difficult to discern the key conclusions and appreciate the utility of the findings. The discussion of subsidies could be more concise. Secondary elements, such as competition intensity and equilibrium results, dilute the main theme.

1. Condense some contextual details to focus on the specific gaps in research related to channel encroachment and government subsidies and include a clear thesis statement, summarizing the research's contribution.

2. Some general findings are repeated without directly connection to the study's objectives. Add a explicit comparison of how this study differs from others.

3. The assumption of linear subsidies and static competition intensity limit the model's applicability to dynamic markets. Incorporating uncertainties like fluctuating biomass prices would enhance the robustness of the model.

While the model can predict outcomes, it could provide some actionable insights for policymakers.

4. The conclusions could link findings to existing studies to highlight the contributions. Including a figure could enhance the presentation of key findings. The conclusions could be condensed to improve readability.

6. PLOS authors have the option to publish the peer review history of their article (what does this mean? ). If published, this will include your full peer review and any attached files.

**Do you want your identity to be public for this peer review?** For information about this choice, including consent withdrawal, please see our Privacy Policy .

Reviewer #1: No

Reviewer #2: No

Reviewer #3: No

---

## [Author Response · Author response to Decision Letter 0]

25 Feb 2025

We have meticulously addressed each suggestion from the academic editor and reviewers in detail. For the detailed responses, please refer to the attached document "Response to Reviewers".

---

## [Decision Letter · Decision Letter 1]

28 Mar 2025

Research on Government Subsidy Strategy of Biomass Power Supply Chain Considering Channel Encroachment

PONE-D-24-49884R1

Dear Dr. Xu,

We’re pleased to inform you that we have reassessed your manuscript and your manuscript has been judged scientifically suitable for publication and will be formally accepted for publication once it meets all outstanding technical requirements.

Kind regards,

Annesha Sil, Ph.D.

Staff Editor

PLOS One

Additional Editor Comments (optional):

Reviewers' comments:

Reviewer's Responses to Questions

**Comments to the Author**

1. If the authors have adequately addressed your comments raised in a previous round of review and you feel that this manuscript is now acceptable for publication, you may indicate that here to bypass the “Comments to the Author” section, enter your conflict of interest statement in the “Confidential to Editor” section, and submit your "Accept" recommendation.

Reviewer #1: All comments have been addressed

Reviewer #2: All comments have been addressed

Reviewer #3: All comments have been addressed

2. Is the manuscript technically sound, and do the data support the conclusions?

Reviewer #1: Yes

Reviewer #2: Yes

Reviewer #3: Yes

3. Has the statistical analysis been performed appropriately and rigorously? 

Reviewer #1: Yes

Reviewer #2: N/A

Reviewer #3: Yes

4. Have the authors made all data underlying the findings in their manuscript fully available?

Reviewer #1: Yes

Reviewer #2: Yes

Reviewer #3: Yes

5. Is the manuscript presented in an intelligible fashion and written in standard English?

Reviewer #1: Yes

Reviewer #2: Yes

Reviewer #3: Yes

6. Review Comments to the Author

Reviewer #1: | appreciate the authors' efforts. I have no further questions. However, the format of this article is strange and should be carefully revised and proofread with reference to the journal's formatting requirements.

Reviewer #2: The authors did good work to revise the manuscript. I would be happy to recommend the publication.

Reviewer #3: (No Response)

7. PLOS authors have the option to publish the peer review history of their article (what does this mean? ). If published, this will include your full peer review and any attached files.

**Do you want your identity to be public for this peer review?** For information about this choice, including consent withdrawal, please see our Privacy Policy .

Reviewer #1: No

Reviewer #2: No

Reviewer #3: **Yes: ** João Zambujal-Oliveira

---

## [Editor Report · Acceptance letter]

PONE-D-24-49884R1

PLOS ONE

Dear Dr. Xu,

I'm pleased to inform you that your manuscript has been deemed suitable for publication in PLOS ONE. Congratulations! Your manuscript is now being handed over to our production team.

Kind regards,

on behalf of

Dr Annesha Sil

Staff Editor

PLOS ONE